An improved termite life cycle optimizer algorithm for global function optimization

Wang Yanjiao
Wei Mengjiao 18437513309@163.com
School of Electrical Engineering, Northeast Electric Power University , Jilin, Jilin , China
Alatas Bilal
Electronic publication date: 2025 Feb 17
Publication date: 2025
Volume: 11
Electronic Location ID: e2671
Received 2024 Oct 28; Accepted 2025 Jan 6
Copyright: © 2025 Wang and Wei
Copyright year: 2025
Copyright holder: Wang and Wei
License: This is an open access article distributed under the terms of the Creative Commons Attribution License, which permits unrestricted use, distribution, reproduction and adaptation in any medium and for any purpose provided that it is properly attributed. For attribution, the original author(s), title, publication source (PeerJ Computer Science) and either DOI or URL of the article must be cited.
License URL: https://creativecommons.org/licenses/by/4.0/

Keywords: Termite life cycle optimizer algorithm, Novel generative strategy, Novel replacement update mechanism, Adaptive crossover, Metaheuristic algorithm.

Funding: Project of Scientific and Technological Innovation Development 20210103090 This work is supported by the Project of Scientific and Technological Innovation Development of Jilin in China under Grant 20210103090. The funders had no role in study design, data collection and analysis, decision to publish, or preparation of the manuscript.

==============================
The termite life cycle optimizer algorithm (TLCO) is a new bionic meta-heuristic algorithm that emulates the natural behavior of termites in their natural habitat. This work presents an improved TLCO (ITLCO) to increase the speed and accuracy of convergence. A novel strategy for worker generation is established to enhance communication between individuals in the worker population and termite population. This strategy would prevent the original worker generation strategy from effectively balancing algorithm convergence and population diversity to reduce the risk of the algorithm in reaching a local optimum. A novel soldier generation strategy is proposed, which incorporates a step factor that adheres to the principles of evolution to further enhance the algorithm’s convergence speed. Furthermore, a novel replacement update mechanism is executed when the new individual is of lower quality than the original individual. This mechanism ensures a balance between the convergence of the algorithm and the diversity of the population. The findings from CEC2013, CEC2019, and CEC2020 test sets indicate that ITLCO exhibits notable benefits regarding convergence speed, accuracy, and stability in comparison with the basic TLCO algorithm and the four most exceptional meta-heuristic algorithms thus far.

Introduction

Meta-heuristic algorithms are currently the most effective methods for solving optimization problems in various fields, which mainly include the following three categories: meta-heuristic algorithms inspired by evolutionary processes, such as genetic algorithms (Kaveh, 2014), genetic programming algorithms (Sette & Boullart, 2001), and biogeography-based optimization algorithms (Simon, 2008), etc.; meta-heuristic algorithms inspired by physics, such as gravitational local search algorithm (Abuhamdah, Ayob & Kendall, 2014), gravitational search algorithm (Rashedi, Rashedi & Nezamabadi-Pour, 2018), small world optimization algorithm (Du, Wu & Zhuang, 2006), light ray optimization algorithm (Kaveh & Khayatazad, 2012), etc.; meta-heuristic algorithms inspired by the behavior of biota groups, e.g., particle swarm optimization algorithm (Kennedy & Eberhart, 1995), difference algorithm (Storn & Price, 1997), gray wolf optimization algorithm (Mirjalili, Mirjalili & Lewis, 2014), termite algorithm (Ajith et al., 2006), monkey search algorithm (Mucherino & Seref, 2007), ant-lion optimization algorithm (Mirjalili, 2015) and so on. Presently, scholarly investigations pertaining to metaheuristic algorithms mostly center around two key domains: enhancing the efficacy of pre-existing metaheuristic algorithms and introducing novel metaheuristic algorithms.

Researchers have made enhancements to current meta-heuristic algorithms, and the following study findings are more representative: In their study, Gupta & Deep (2019) introduced the improved sine cosine algorithm. This algorithm demonstrates enhanced capabilities in exploring the search space, mitigating local optimality, achieving effective convergence for global optimality, and optimizing the search space. Xu, Wan & Huang (2011) proposed an improved optimized genetic algorithm, which balances the exploration and exploitation capabilities while enhancing the population diversity; Wang & Liu (2023) proposed the improved elephant clan optimization (IECO) Algorithm, which enhances the population diversity by attaching an autonomous movement strategy for population initialization, a Euclidean distance-based community partitioning, and a precocious suppression mechanism; Wang, Chen & Ku (2023) proposed an improved archimedes optimization (IAOA) Algorithm, which introduces a simple method mechanism to correct for poorer individuals and enhance the algorithm’s convergence speed and accuracy; Muhammed, Saeed & Rashid (2020) proposed the improved FDO (IFDO) algorithm to augment the fitness-dependent optimizer (FDO) in the exploratory and exploitation phases by considering. Deng & Liu (2023) introduced a multi-strategy improved slime mold algorithm (MSMA) in order to balance the exploitative and exploratory features of traditional metaheuristic algorithms (MAs), Yao et al. (2023) proposed an enhanced snake optimizer (ESO) by introducing a new dyadic-based learning strategy and a new dynamic updating mechanism to obtain better performance;To balance exploration and exploitation is the key to enhancing algorithm performance and problem adaptability, Jia & Lu (2024) proposes a brand new strategy named Guided Learning Strategy (GLS) to solve above problem.

Concurrently, a substantial quantity of innovative meta-heuristic algorithms have been suggested. Abdullah & Ahmed (2019) proposed the fitness-dependent optimizer (FDO), which was inspired by the swarming of honeybees during their reproduction process and their collective decision-making behaviors; and Sharma & Ghosh (2020) proposed the Jackson’s Widowbird Mating Optimization (JWMO) by simulating courtship behaviors carried out by male birds; Talatahari et al. (2021) proposed the crystal structure algorithm (Crystal), which was mainly inspired by the basic principle of adding radicals to lattice sites to form crystal structures; Trojovský, Dehghani & Hanuš (2022) proposed the Siberian Tiger Optimization Algorithm, which was basically inspired to mimic the natural behaviors of northeastern tigers during hunting and fighting; The Orchard Algorithm (OA) is designed and introduced by Kaveh, Mesgari & Saeidian (2023), inspired by fruit gardening, it can efficiently search and explore the search space; Ghiaskar, Amiri & Mirjalili (2024) proposed article introduces a new optimization algorithm inspired by nature called the polar fox optimization algorithm (PFA); (Ghasemi et al., 2024) introduce a novel meta-heuristic optimization algorithm, the Flood Algorithm (FLA) draws inspiration from the intricate movement and flow patterns of water masses during flooding events in river basins. Minh et al. (2023) introduced a novel bionic meta-heuristic optimization algorithm called the termite life cycle optimizer (TLCO) in the same year. TLCO emulates the natural behavior of termites in their natural habitat.

New metaheuristic algorithms exhibit more pronounced advantages in terms of convergence speed, robustness, and parallel computing capacity than standard metaheuristic algorithms. Hence, enhancing the efficacy of novel metaheuristic algorithms has emerged as a prominent area of research within the field of evolutionary science. The termite life cycle TLCO algorithm has the advantages of minimal parameter setting and parallel computation. A number of experiments have proved that the convergence speed and accuracy of the TLCO algorithm are better than gravitational search algorithm (GSA), grey wolf optimizer (GWO), Cuckoo search (CS), whale optimization algorithm (WOA), sine cosine algorithm (SCA), moth-flame optimization algorithm (MFO), Harris Hawks Optimizer (HHO) and arithmetic optimization algorithm (AOA). In this regard, the present study aims to enhance the TLCO algorithm to obtain high convergence speed and convergence accuracy in practical applications. The primary advancements and driving forces are as follows:

1) Strategy for generating new workers: various updating techniques are selected based on the effectiveness of individual adaptation while enhancing communication with other members of the termite population by learning from the optimal individual. This approach prevents the original worker generation strategy from effectively balancing algorithm convergence and population diversity, thereby reducing the risk of the algorithm in reaching a local optimum.

2) Strategy for the next generation of soldiers: the proposed approach involves selecting the optimal individual or a subset of superior individuals as the base vector. Random vectors that adhere to the principles of evolution and exhibit dissimilarity across all dimensions are introduced as the step factor. This approach aims to address the issue of the original algorithm having an equal number of moving steps in each dimension while also enhancing the convergence speed of the algorithm.

3) Implementation of a novel mechanism for updating replacements: an adaptive crossover factor is implemented when the new individual is of lower quality than the original individual. This factor facilitates the crossover between the original individual and the corresponding individual in the termite population. It prevents the original algorithm from directly replacing the original individual with the new individual. Additionally, it ensures a balance between the convergence of the algorithm and the diversity of the population. When evaluated on the CEC2013, CEC2019 and CEC2020 test set, the findings indicate that the ITLCO algorithm, as suggested in this study, outperforms TLCO and four other optimization algorithms in terms of convergence speed, convergence accuracy, and stability.

This article is structured in the following manner: “Termite Life Cycle Optimizer Algorithm” provides an explanation of the operational principle and procedure of the TLCO algorithm. In “Improved Termite Life Cycle Optimizer Algorithm”, an analysis is followed by the proposal of an enhanced ITLCO algorithm. The simulation results and analysis are presented in “Experiment”. “Conclusions” provides a concise overview of the algorithm presented in this research.

Termite life cycle optimizer algorithm

Termites are widely distributed social insects. A termite colony consists of a termite queen, a large number of worker termites, soldier termites, and reproductive termites. Among them, worker termites, soldier termites, and reproductive termites take on different tasks and possess a high level of social organization. Worker termites are responsible for tasks such as foraging and nest-building, while soldiers are responsible for protecting the colony. However, when termite workers fail to find a better colony or soldiers fail to protect it, the reproductive termite needs to find a new source of food and create a new colony. Finding a good colony to develop is the goal of the termite population. Minh et al. (2023) introduced a novel algorithm called the termite life cycle optimizer algorithm, which was influenced by the behavior of termites. TLCO involves the construction of four distinct populations, including the termite population, worker population, soldier population, and reproduction population. In the present scenario, the populations of workers and soldiers are utilized to simulate the behaviors exhibited by workers and soldiers, respectively. The total count of termite individuals within these populations is equivalent to the combined count of termite individuals within the termite population and the reproduction population. The operations for the worker stage, soldier stage, and reproduction stage are specifically tailored for the worker, soldier, and reproduction populations, respectively. Algorithm 1 displays the pseudo-code of TLCO, with a concise description of the major steps.

Algorithm 1 The pseudo-code of TLCO.

Input: N: population size; D: optimizing problem dimensions; Kmax: maximum number of iterations; trial i: unrenewed marker sites for individual termites	
Output: optimal solution and its fitness value	
01. Initialize parameters (N, D, Kmax)	
02. Randomly spawn an initial termite population X in the domain of definition and initialize the unupdated flag bits of termite individuals’ triali	
03. Calculate the fitness value f (Xi) for each individual Xi	
04. Determine initial worker population Xworker←X (1:0.7N, :)	
05. Determine initial soldier population Xsoldier←X (0.7N+1: N, :)	
06. Determine initial reproductive population Xreproductive←X	
07. For k = 1: Kmax do	
              // worker phase	
08.    For each termite Xworker in the worker population Xworker,i do	
09.      Xworker,i←Xworker,i update worker as described in “Worker Phase”	
10.            if f (Xworker,i) < f (Xi) then	
11.              X i←Xworker,i	
12.            else	
13.              triali =trial i+1	
14.            end if	
15.    End For	
            // soldier phase	
16.    For each termite Xsoldier in the soldier population Xsoldier,i do	
17.      Xsoldier,i←Xsoldier,i update soldier as described in “Soldier Phase”	
18.            if f (Xsoldier,i) < f (Xi) then	
19.              X i+0.7N←Xsoldier,i	
20.            else	
21.              triali+0.7N =triali + 0.7N+1	
22.            end if	
23.    End For	
24.    Update reproductive as described in “Reproductive phase”	
25.    k=k + 1	
26. End For	
27. Output the global optimal solution	

Worker phase

Termites primarily participate in activities such as food search, shelter construction, and reproduction within their natural habitat. The TLCO algorithm is designed to imitate the aforementioned characteristics by implementing the worker update operation, as depicted in Eq. (1), for termite workers throughout the worker population. This process involves a continual exploration of new search space.

(1) Xworker,i(k+1)=Xworker,i(k)+ξ×(S∗(k+1)+rand(1,D))×|Xbest(k)−Xworker,i(k)|.

In this context, k denotes the current iteration number, D represents the dimension of the optimization problem, Xbest(k) represents the current best individual of the termite population, ξ is a scalar that controls the moving direction of the termite workers, and its value is limited to the range of [−1, 1], as indicated in Eq. (2), and S∗(k+1) represents the Levy flight step, as indicated in Eq. (3).

(2) ξ=−1+rand×2

(3) S∗k=U|V|1/β.

The symbol “ |∙|” represents the operation of calculating the absolute value. The variable “ β” represents the Levy distribution index, which is limited to a range of Xbest(k), as indicated in Eq. (4). The variables “ U” and “ V” correspond to the normal distribution, with values of U(≈⁡0,σu2) and V(≈⁡0,1), respectively. The variance of these variables is σu, as shown in Eq. (5).

(4) β=1.5+0.5kKmax

The maximum number of iterations is Kmax.

(5) σu={Γ(1+β)sin⁡(πβ/2)Γ[(1+β)/2]β2(β−1)/2}1/β.

When the gamma function Γ(∙) is applied to the real numbers z, the resulting value can be represented as Eq. (6).

(6) Γ(z)=∫0∞tz−1e−tdt.

It is noteworthy to mention that the workers within the worker population undergo direct updates as per Eq. (1), hence obviating the necessity of comparing the adaptation values of the workers prior to and subsequent to the update. If the updated worker outperforms the corresponding termite individual in the termite population, the corresponding termite individual is replaced. Otherwise, the un-updated flag bit triali of the corresponding termite individual is incremented by 1.

Soldier phase

Termite soldiers, in nature, only relocate in close proximity to their colonies in order to safeguard the termite queen Xbest. The soldier update procedure for termite soldiers Xsoldier,i in the soldier population Xsoldier is designed by the TLCO algorithm, as depicted in Eq. (7), in order to replicate the aforementioned behavior.

(7) Xsoldier,i(k+1)=2×rand×Xsoldier,i(k)+(−1+2×rand)×|Xsoldier,i(k)−S∗(k+1)×Xbest(k)|.

The Levy flight step, denoted as S∗(k+1), is determined using Eq. (3).

It is important to highlight that the soldiers within the soldier population undergo direct updates as per Eq. (7), hence obviating the necessity of comparing the adaption values of the soldiers prior to and subsequent to the update. In the event that the updated soldier surpasses the matching termite individual within the termite population, the former is substituted. Conversely, if the latter is not improved, the un-updated flag bit triali of the former is incremented by 1.

Reproductive phase

When the unrenewed marker bit triali of the i termite individual satisfies Eq. (8), the corresponding termite individual in the reproducing population performs the reproductive operation as described by Eq. (9) and returns to a value of triali to 0. The calculation of the probability λi, which controls the probability of the occurrence of breeding termites, is performed using Eq. (10).

(8) triali>Kmax×λi

(9) Xreproductive,i(k+1)={Xr1(k)+rand(1,D)×(Xr2(k)−Xr1(k)),iff(Xr2(k))<f(Xr1(k))Xr1(k)−rand(1,D)×(Xr2(k)−Xr1(k)),else

(10) λi={1−11+e−σ(k−0.5Kmax),ifi≤0.7N11+e−σ(k−0.5Kmax),else

where r1,r2∈(1,2,…,N) and r1≠r2≠i. where the constant σ is equal to 1/0.1Kmax.

Following the completion of the breeding operation, it is observed that all individuals within the breeding population undergo renewal. Additionally, individuals within the remaining population may also undergo renewal in the following manner: if the fitness of an individual Xreproductive,i surpasses that of the corresponding individual Xi in the termite population, it is directly substituted by an individual Xreproductive,i. According to Eq. (11), the individual Xreproductive,i in the reproductive population will undergo re-updating. Simultaneously, the outcome of this updating process will be directly substituted with either the worker individual Xworker,i in the worker population or the soldier individual Xsoldier,i−0.7N in the soldier population.

(11) Xreproductive,i(k+1)=Lb+rand(1,D)(Ub−Lb)

where Lb and Ub are the lower and upper bounds of the domain of definition, respectively.

Improved termite life cycle optimizer algorithm

Starting the reproduction stage requires triali to satisfy Eq. (8). For instance, when considering the first 0.7N individuals in the reproducing population, as depicted in Eq. (10), the value of λi decreases as the number of iterations increases. In the early iteration period, it is approximately 0.9931, indicating that individuals must accumulate up to 1,976 function evaluations without updating to enter the reproduction stage. In the later iteration period, the value decreases to approximately 0.0069, requiring individuals to accumulate up to 14 function evaluations without updating to enter the reproduction stage. The initial 0.7N individuals within the reproductive population exhibit a diminished likelihood of engaging in the reproductive process, and a similar situation is observed among the final 0.3N individuals. In the replacement update mechanism of Algorithm 1, the likelihood of substituting the original worker/soldier individual with the individual produced through the reproduction operation is minimal, whereas the likelihood of directly substituting the original worker/soldier individual with the new worker/soldier individual is substantial. In summary, the TLCO algorithm primarily focuses on executing the worker phase, soldier phase, and direct replacement update mechanism. Conversely, the replacement update is less frequently encountered during the reproduction phase. Hence, the improved termite life cycle optimizer algorithm (ITLCO) is proposed as an alternative to enhance the convergence performance of the TLCO method. This algorithm primarily focuses on enhancing the worker phase, soldier phase, and direct replacement updating mechanism.

Strategies for the generation of new workers

An in-depth analysis of Eq. (1) shows that each worker within the termite population exclusively acquires knowledge from the current optimal individual. This phenomenon has the potential to expedite the convergence of the algorithm. However, relying solely on a single learning object may result in individuals becoming excessively similar. Moreover, if the current optimal individual is locally optimal, a significant risk of the population being trapped in a local optimum exists. The difference vector component can be understood as the multiplication of the step size and |Xbest(k)−Xworker,i(k)|. Given that |Xbest(k)−Xworker,i(k)| is positive in every dimension, the difference vector component is either positive or negative in each dimension. This finding implies that each dimension of the individual worker either increases or decreases. In the case of a D-dimensional solution space, approximately 2D directions exist, and any increases or decreases in each dimension are limited to only two directions within the solution space. This phenomenon significantly diminishes the population diversity and significantly heightens the likelihood of the algorithm in encountering a local optimum. In addition, its step size is relatively random. Although it can increase the diversity of the population, ensuring that the individual migrates to the excellent position is difficult because of excessive blindness, thereby slowing down the convergence speed of the algorithm. In conclusion, the initial approach to worker creation lacks the ability to adequately reconcile the convergence of the algorithm with the diversity of the population.

To achieve a harmonious equilibrium between algorithm convergence and population diversity, a novel worker update technique is introduced, as depicted in Eq. (12).

(12) Xworker,i(k+1)={Xworker,i(k)+(S∗(k+1)+rand(1,D))×(Xbest(k)−Xworker,i(k))+rand(1,D)×(Xr1(k)−Xr2(k)),iff(Xworker,i(k))<f(Xsort(m)(k))mean(Xworker(k))+rand(1,D)×(Xbest(k)−Xr1(k))+rand(1,D)×(Xr2(k)−Xworker,i(k)),else.

Here, r1 and r2 are two distinct individuals chosen at random from the termite population, namely r1,r2∈(1,2,…,N) and r1≠r2≠i. The variable Xsort(m)(k) represents the m-th individual after sorting the worker population from smallest to largest based on fitness at the kth iteration. In general, when N = 50, m = 30 can yield a superior outcome, which can also be adjusted based on the optimization problem.

Compared with the worker generation strategy of the original algorithm, the new worker generation strategy proposed in this section has the following innovations. First, for the base vector, the basic TLCO algorithm only takes the base vector, while the ITLCO algorithm takes or as the base vector according to the merits and disadvantages of individuals. Second, the basic TLCO algorithm only learns from others, while the ITLCO algorithm not only learns from others but also introduces the communication between other individuals. Third, the ITLCO algorithm changes the search step size of the TLCO algorithm. The worker generation technique provided in this section offers some advantages when compared with the worker generation strategy of the original algorithm. In the innovative worker generation strategy, individuals no longer search for near themselves or the population, in contrast to the original update approach. Given that the least favorable individuals can offer a certain degree of population diversity, their impact on the convergence of the algorithm is notably limited. Conversely, the central region of the worker population is expected to possess superior fitness values. The exploration of this central region is expected to expedite the convergence of the algorithm. Furthermore, the central region of the worker population exhibits distinct genotypic characteristics compared with the remaining workers. This property ensures that the reduction in population diversity caused by the elimination of the least favorable individuals can be offset, thereby avoiding an excessive reduction in population diversity. The potential impact on population diversity is not expected to be significant. The innovative worker generation approach is a distinct update strategy that has been developed for multiple individuals with the lowest individual fitness. This strategy aims to strike a balance between algorithm convergence and population diversity to some degree. In the novel worker generation strategy, the individual workers not only acquire knowledge from the optimal individuals but also enhance their communication with other individuals within the termite population. This approach enhances the algorithm’s global search capability and mitigates the drawback of the original generation strategy, which is susceptible to local optima when relying solely on learning from the optimal individuals. The random scalar factor ξ in the initial step factor ξ×(S∗(k+1)+rand(1,D)) is eliminated, leaving only the random perturbation component derived from the Levy flight step. This modification partially mitigates the algorithm’s blindness while maintaining the population’s diversity. Furthermore, the absolute value of |Xbest(k)−Xworker,i(k)| is eliminated, and the information between an individual and the optimal individual is employed to automatically ascertain its movement direction in the search space. This step results in a greater number of moving directions compared with ξ, which can only be increased or decreased in a unidirectional manner. Consequently, the population diversity is further enhanced. In brief, the worker generation technique suggested in this section offers a novel approach that effectively addresses the requirements of population variety and convergence speed.

To visually examine how the new worker generation strategy in this section can effectively balance the convergence rate and population diversity of the algorithm, we make the following comparison. We replace the worker strategy in the TLCO algorithm with the new worker generation strategy proposed in this section to form a new algorithm, called ITLCO_worker. The ITLCO_worker algorithm and the TLCO algorithm are tested on the F1 and F6 of the CEC2013 test set to ensure that the initial populations corresponding to them are identical. The other parameters are set as follows: the number of populations is 50, and the dimension of the optimization problem is 2. Table 1 shows the corresponding population scatter distribution map and the corresponding function evaluation times when the accuracy of the optimal solutions of the two algorithms reaches 10−3, 10−6, and 10−9.

Table 1 Comparison of population distribution.

	F1	F6	
	TLCO	ITLCO_worker	TLCO	ITLCO_worker	
10−3					
10−6					
10−9					

As shown in Table 1, when the same convergence accuracy is achieved, the individual distribution of the ITLCO_worker is significantly more dispersed than that of the ITLCO algorithm, and fewer times of function evaluation are required. Therefore, the new worker search strategy proposed in this section can balance the convergence rate and population diversity.

Strategies for the generation of new soldiers

An in-depth analysis of Eq. (6) can be seen. The basis vector 2×rand×Xsoldier,i(k) induces the random migration of all dimensions of the original soldier within the range of [0,2] close proximity to itself. The difference vector |Xsoldier,i(k)−S∗(k+1)×Xbest(k)| further facilitates the migration of the migrated soldier towards the global optimal position. The inclusion of the difference vector component enhances the algorithm’s convergence rate, while the inclusion of the base vector component contributes to increased population variety. Nevertheless, the phenomenon of random migration in all dimensions can lead to the disruption of the soldier’s original evolutionary trajectory. Consequently, a high probability that the soldier will migrate to a suboptimal position exists. Even if a subsequent migration to the global optimum position is executed, the likelihood of surpassing the original soldier’s performance is significantly diminished. In conclusion, the TLCO algorithm is specifically designed to generate soldiers in a manner that ensures a certain degree of population diversity, albeit potentially impeding the algorithm’s convergence speed.

This section presents a new soldier generation strategy, as depicted in Eq. (13), which aims to achieve a balanced convergence and diversity of the algorithm.

(13) Xsoldier,i(k+1)={Xg+b×(Xsoldier,i(k)−Xbest(k)),ifrand<0.5Xg+b×(Xr1(k)−Xr2(k)),else

where, Xg is the better individual, which is determined as shown in Eq. (14); b is the adaptive step factor, which is formulated as shown in Eq. (15); r1, r2 are two different individuals randomly selected in the termite population, i.e., r1,r2∈(1,2,…,N) and r1≠r2≠i.

(14) Xg(k)={Xpbest(k),ifrand<0.5Xbest(k),else

where Xpbest(k) represents a randomly selected individual from the optimal p individuals in the termite population. Where p is taken as five will generally give better results or can be set on its own according to the optimization problem.

(15) b=(2−2×k/Kmax)×rand(1,D)

In conclusion, by comparing Eq. (7) with Eq. (13), the new soldier generation strategy proposed in this section has the following improvements compared with the update strategy of the soldier stage in the basic TLCO algorithm. First, the basic TLCO algorithm takes the individual itself as the base vector, and we choose the more excellent individual as the base vector. Second, in the basic TLCO algorithm, individuals only learn from the best individuals, while ITLCO also introduces the opportunity for individuals to communicate with each other. Third, ITLCO adaptively controls the search step size. In-depth analysis shows that the innovative troop generation approach possesses the subsequent benefits. In contrast to the blind migration method used to obtain basis vectors from the initial individuals, the novel soldier generation strategy employs a random selection process from either the optimal individuals or a few superior individuals with equal chance. This aspect of methodology exhibits a conspicuous influence of exceptional data, hence expediting the algorithm’s convergence rate. In addition, the selection of basis vectors is not unique, which maintains population diversity to a certain extent. Second, in the novel soldier generation strategy, the original worker stage is replicated. However, instead of solely relying on the global optimal individual, the strategy incorporates learning from other individuals through mutual communication. This approach ensures faster convergence, enhances population diversity, and reduces the risk of the algorithm falling into a local optimum. The initial method of learning from the ideal global individual involves assigning an equal random number to each move step on each dimension. However, the degree of evolution of individuals in each dimension of space is not uniform. Consequently, this approach is clearly unsuitable. The new soldier generation strategy, as described in Eq. (15), allows for global optimal individual learning and inter-individual learning. Each dimension of the move step is not assigned a random number, resulting in increased population diversity. Furthermore, as the number of iterations increases, the value of the move step gradually decreases. This approach aligns with the law of evolution, which states that evolution requires a wide range of rapid exploration in the pre-evolutionary period. This finding aligns well with the principles of evolution, which dictate that during the early stages of development, quick exploration throughout a wide range is necessary. In the later stages, a more precise search is conducted to prevent the theoretical optimum from being overlooked due to the extensive search range. In contrast to the initial generation strategy, the novel soldier generation strategy introduced in this part employs two distinct generation methods concurrently, thereby increasing the algorithm’s capacity to address various optimization challenges to a certain degree.

Novel mechanism for replacement renewal

According to Algorithm 1, individuals within the termite population will undergo update under two circumstances: first, if the newly generated individuals Xworker,i and Xsoldier,i in the worker and soldier populations outperform their counterparts Xi in the termite population; and second, if the newly generated individuals Xreproductive,i in the reproduction population outperform their counterparts in the termite population. In the first scenario, the newly formed individuals within the worker and soldier populations will directly supplant the original individuals, regardless of their inferiority. Therefore, the corresponding individuals in the termite population are not inferior to those in the worker and soldier populations. In the second scenario, if the newly produced individuals from the reproducing population exhibit superior traits compared with their counterparts in the termite population, the termite population will be substituted by the newly generated individuals. However, the corresponding individuals in the worker and soldier populations are unchanged. Individuals in the termite population exhibit superiority over their counterparts in the worker and soldier populations. When the new individuals produced by the reproducing population are inferior to their counterparts in the termite population, new individuals will be randomly generated within the defined domain to replace the counterparts in the worker and soldier populations, while the counterparts in the termite population remain unaltered. The probability of surpassing the appropriate individual in the termite population is exceedingly low because of the significant blindness associated with the randomly created position. Thus, for case two, individuals in the termite population are almost always superior to their counterparts in the worker and soldier populations. In conclusion, among the termite population, individuals tend to exhibit superior performance compared with their counterparts in the worker and soldier populations as the iteration advances.

The utilization of the worker and soldier population update procedures in “Worker Phase” and “Soldier Phase” reveals the potential for enhancing population variety by directly replacing the original individuals with the new individuals created. Nevertheless, if the new individuals are of lower quality than the original people, then direct substitution will surely decrease the algorithm’s convergence speed. A novel update replacement operation is proposed in Eq. (16) to enhance the convergence speed of the algorithm and enhance the diversity of the population. This operation involves performing a dimension-by-dimension crossover operation between the original worker or soldier individual and the corresponding individual in the termite population when the new worker or soldier individual is inferior to the original individual.

(16) Xpop,(i,j)(k)={Xi,j(k),ifrand<CRorj=jrandXpop,(i,j)(k),else

where XPOP represents either the worker population or the soldier population and jrand is a random number in {1,2,…,D}; the crossover factor CR is shown in Eq. (17).

(17) CR=0.8−0.3×k/Kmax.

In summary, CR∈[0.5,0.8] and gradually decrease with the increase of iterations. The genes of new individual workers or soldiers are predominantly derived from termite populations. During the early iteration period, the genes of new individual workers or soldiers have a higher proportion of termite populations, which improves the convergence speed of the algorithm. However, during the later stages of iteration, population diversity is maintained while the convergence speed is enhanced.

Algorithm complexity analysis

Suppose that the population size of ITLCO algorithm is N, where the number of worker population and soldier population are Nw = 0.7 N and Ns = 0.3 N, respectively, and the problem dimension is D. ITLCO algorithm mainly includes three update strategies: worker stage, soldier stage and reproduction stage. The worst-time complexity analysis of each stage of the ITLCO algorithm is as follows. In the worker stage, the Nw times Eq. (12) needs to be calculated at most, which includes two multiplications and five additions. The worst-time complexity of this stage is O(2 × Nw × D + 5 × Nw). In the soldier stage, it is necessary to calculate at most the Ns times Eq. (13), which includes two multiplications and two additions. Hence, the worst-time complexity of this stage is O(2 × Ns × D + 2 × Ns). At most N times, Eq. (9) needs to be calculated in the breeding stage, which consists of one multiplication and two additions. The worst-time complexity of this stage is O(1 × N × D + 2 × N).

ITLCO, therefore, a single operation for the worst of time complexity is O(2 × Nw × D + 5 × Nw) + O(2 × Ns × D + 2 × Ns) + O(1 × N × D + 2 × N) ≈ O(3 × N × D + 6 × N).

Experiment

To assess the performance of the ITLCO algorithm, this section presents experiments divided into two parts: (1) verifying the effectiveness of the proposed three improvement strategies and (2) comparing the performance of the ITLCO algorithm against the basic TLCO algorithm as well as four other more prominent evolutionary algorithms known for their effectiveness.

This section utilizes CEC2013, CEC2019, and CEC2020 test sets for experimental simulations. The CEC2013 test set (Liang et al., 2013) contains 28 test functions, of which F1–F5 are unimodal functions, which have only one optimal value and are used to verify the convergence performance of the algorithm. F6–F20 is a multi-peak function, which has multiple local optimal solutions and is used to verify the ability of the algorithm to escape the local optimal. F21–F28 are compound functions. The CEC2019 test set contains 10 test functions, and F1–F10 are multi-peak functions. The CEC2020 test set contains 10 test functions; among which, F1 is a unimodal function, F2–F4 is a multi-modal function, F5–F7 is a mixed function, and F8–F10 is a compound function.

All experiments were conducted on a computer running Windows 11, CPU 13th Gen Intel(R) Core (TM) i9-13900H CPU @ 2.60 GHz, and implemented using MATLAB R2020b.

Empirical investigations on the efficacy of each enhancement strategy

The ITLCO algorithm enhances the TLCO algorithm through three distinct methods. In this study, we evaluate the efficacy of these three improvement strategies by eliminating one of the existing improvement strategies in ITLCO. The three new improvement algorithms: the new worker generation strategy with “Strategies for the Generation of New Workers” removed from ITLCO, the new soldier generation strategy with “Strategies for the Generation of New Soldiers” removed from ITLCO, and an improved algorithm with a novel replacement update mechanism with “Novel Mechanism for Replacement Renewal” removed from ITLCO. These algorithms are then compared with the ITLCO algorithm using the CEC2013 test set. To facilitate comprehension, the aforementioned three enhanced algorithms are designated as ITCO1, ITCO2, and ITCO3, correspondingly.

To maintain fairness in the comparison, each method has a population size of N = 50, a test function with a dimensionality of D = 30, and a maximum limit of 100,000 function evaluations, denoted as MaxFEs. To mitigate the risk of a singular algorithm operation, the algorithms are executed autonomously for a total of 30 iterations on each test function.

The running results of each algorithm on the 30-dimensional CEC2013 function set are presented in Table 2. The variables “±” before and after representing the mean and standard deviation of the optimal values in 30 experiments, respectively. To further confirm the distinctions between the ITLCO algorithm and the enhanced algorithms, we conducted the Wilcoxon rank sum test (Derrac et al., 2011) with a significance level of 5% and performed Friedman’s test. The specific outcomes of these tests are presented in Tables 3 and 4. Wilcoxon rank-sum test, proposed by Frank Wilcoxon in 1945, is a non-parametric hypothesis testing method, which is generally used to detect the difference between non-paired data. In the context of statistical analysis, a p-value greater than 0.05 signifies the absence of a significant difference between the improved algorithm and the ITLCO algorithm. This is denoted by the symbol “=”. Conversely, if the p-value is less than 0.05 and the average of the optimal solutions obtained from 30 experiments for the improved algorithm surpasses that of the ITLCO algorithm, it indicates a significant superiority of the improved algorithm over the ITLCO algorithm. When the p-value is below 0.05 and the mean value of the optimal solutions obtained by the improved algorithm in 30 experiments surpasses that of the ITLCO algorithm, it signifies that the improved algorithm is clearly superior to the ITLCO algorithm, denoted by the symbol “+”. Conversely, if the p-value is less than 0.05, it indicates that the performance of the improved algorithm is significantly inferior to that of the ITLCO algorithm, denoted by the symbol “−”. The Friedman test was proposed by M. Friedman in 1973 and is used to calculate the comprehensive ranking of various algorithms. The smaller the corresponding rank mean of the algorithm, the higher the ranking of the algorithm, indicating the better the overall performance of the algorithm.

Table 2 Operational results of each improvement strategy in the 30-dimensional CEC2013 test set.

	ITLCO	ITLCO1	ITLCO2	ITLCO3	
F1	0.00E+00 ± 0.00E+00	1.65E−24 ± 2.57E−24	6.89E−05 ± 3.80E−05	3.04E−29 ± 7.18E−29	
F2	7.73E+05 ± 4.94E+05	4.80E+06 ± 2.81E+06	5.92E+06 ± 2.37E+06	3.90E+05 ± 1.39E+05	
F3	0.00E+00 ± 0.00E+00	0.00E+00 ± 0.00E+00	0.00E+00 ± 0.00E+00	0.00E+00 ± 0.00E+00	
F4	9.15E+03 ± 2.76E+03	1.80E+04 ± 4.88E+03	1.83E+04 ± 4.21E+03	5.67E+03 ± 2.00E+03	
F5	0.00E+00 ± 0.00E+00	5.11E−14 ± 4.77E-14	8.25E−04 ± 3.74E-04	1.48E−26 ± 7.98E-26	
F6	1.36E+01 ± 1.63E+01	4.84E+01 ± 2.69E+01	4.01E+01 ± 2.60E+01	2.80E+01 ± 2.51E+01	
F7	0.00E+00 ± 0.00E+00	0.00E+00 ± 0.00E+00	0.00E+00 ± 0.00E+00	0.00E+00 ± 0.00E+00	
F8	2.10E+01 ± 6.24E−02	2.10E+01 ± 5.24E−02	2.10E+01 ± 6.19E−02	2.10E+01 ± 8.60E−02	
F9	0.00E+00 ± 0.00E+00	0.00E+00 ± 0.00E+00	0.00E+00 ± 0.00E+00	0.00E+00 ± 0.00E+00	
F10	7.55E−02 ± 4.28E−02	9.13E−02 ± 7.07E−02	8.59E−01 ± 2.09E−01	9.80E−02 ± 5.60E−02	
F11	0.00E+00 ± 0.00E+00	0.00E+00 ± 0.00E+00	0.00E+00 ± 0.00E+00	0.00E+00 ± 0.00E+00	
F12	0.00E+00 ± 0.00E+00	1.48E+01 ± 4.46E+01	0.00E+00 ± 0.00E+00	2.55E+01 ± 5.22E+01	
F13	0.00E+00 ± 0.00E+00	0.00E+00 ± 0.00E+00	0.00E+00 ± 0.00E+00	5.33E+00 ± 2.87E+01	
F14	9.09E+02 ± 2.26E+02	7.00E+02 ± 2.50E+02	5.14E+03 ± 1.58E+03	1.59E+03 ± 4.83E+02	
F15	4.27E+03 ± 7.27E+02	4.32E+03 ± 6.51E+02	7.06E+03 ± 4.14E+02	4.24E+03 ± 8.85E+02	
F16	1.49E+00 ± 4.01E−01	1.51E+00 ± 5.20E−01	2.63E+00 ± 3.57E−01	1.35E+00 ± 4.81E−01	
F17	3.81E+01 ± 8.99E+00	4.63E+01 ± 1.18E+01	1.81E+02 ± 2.22E+01	7.52E+01 ± 1.98E+01	
F18	9.91E+01 ± 3.04E+01	1.30E+02 ± 3.80E+01	2.22E+02 ± 2.16E+01	1.02E+02 ± 4.00E+01	
F19	5.63E+00 ± 2.21E+00	8.30E+00 ± 2.29E+00	1.61E+01 ± 1.62E+00	5.89E+00 ± 2.31E+00	
F20	0.00E+00 ± 0.00E+00	2.67E+00 ± 3.82E+00	2.75E-01 ± 1.48E+00	2.08E+00 ± 3.51E+00	
F21	4.00E+02 ± 0.00E+00	4.00E+02 ± 5.28E−13	4.00E+02 ± 5.21E−04	4.00E+02 ± 5.12E−13	
F22	8.71E+02 ± 3.07E+02	5.65E+02 ± 3.17E+02	4.97E+03 ± 1.56E+03	1.51E+03 ± 6.66E+02	
F23	4.85E+03 ± 6.79E+02	4.74E+03 ± 7.91E+02	7.02E+03 ± 1.00E+03	4.48E+03 ± 7.01E+02	
F24	2.00E+02 ± 4.14E−02	2.00E+02 ± 4.58E−02	2.00E+02 ± 2.98E−02	2.00E+02 ± 8.03E−02	
F25	2.32E+02 ± 2.18E+01	2.54E+02 ± 3.89E+01	2.28E+02 ± 3.25E+01	2.44E+02 ± 3.22E+01	
F26	2.93E+02 ± 2.49E+01	2.83E+02 ± 3.72E+01	3.00E+02 ± 8.43E−06	2.97E+02 ± 1.79E+01	
F27	3.15E+02 ± 9.32E-01	3.16E+02 ± 1.13E+00	3.16E+02 ± 1.19E+00	3.18E+02 ± 2.07E+00	
F28	8.75E+02 ± 2.09E+01	9.35E+02 ± 1.65E+01	8.58E+02 ± 1.41E+02	9.00E+02 ± 1.75E+01	

Table 3 Wilcoxon rank sum test results for each improvement strategy and ITLCO.

Function	p-value (vs. ITLCO)	
ITLCO1	ITLCO2	ITLCO3	
F1	0.000(−)	0.000(−)	0.001(−)	
F2	0.000(−)	0.000(−)	0.000(+)	
F3	1.000(=)	1.000(=)	1.000(=)	
F4	0.000(−)	0.000(−)	0.000(+)	
F5	0.000(+)	0.000(−)	0.000(−)	
F6	0.000(−)	0.000(−)	0.000(−)	
F7	1.000(=)	1.000(=)	1.000(=)	
F8	0.210(=)	0.044(+)	0.821(=)	
F9	1.000(=)	1.000(=)	1.000(=)	
F10	0.684(=)	0.000(−)	0.178(=)	
F11	1.000(=)	1.000(=)	1.000(=)	
F12	0.082(=)	1.000(=)	0.011(−)	
F13	1.000(=)	1.000(=)	0.334(=)	
F14	0.004(+)	0.000(−)	0.000(−)	
F15	0.900(=)	0.000(−)	0.900(=)	
F16	0.756(=)	0.000(−)	0.217(=)	
F17	0.007(−)	0.000(−)	0.000(−)	
F18	0.002(−)	0.000(−)	0.819(−)	
F19	0.000(−)	0.000(−)	0.652(=)	
F20	0.000(−)	0.334(=)	0.003(−)	
F21	1.000(=)	1.000(=)	1.000(=)	
F22	0.000(+)	0.000(−)	0.000(−)	
F23	0.641(=)	0.000(−)	0.058(=)	
F24	1.000(=)	1.000(=)	0.021(−)	
F25	0.015(+)	0.454(=)	0.006(−)	
F26	0.237(=)	0.161(=)	0.570(=)	
F27	0.000(+)	0.008(−)	0.000(−)	
F28	0.000(−)	0.066(=)	0.000(−)	
+/=/−	5/14/9	1/12/15	2/13/13	

Table 4 Friedman test results for the four algorithms.

	ITLCO	ITLCO1	ITLCO2	ITLCO3	
Avg.rank	1.82	2.64	3.09	2.45	
Sort	1	3	4	2	

Based on the provided data, the results of the Wilcoxon rank sum test clearly indicate that the ITLCO1 algorithm significantly outperforms the ITLCO algorithm across five test functions. Conversely, the Wilcoxon rank sum test results for the nine test functions are significantly negative, indicating that the ITLCO1 algorithm outperforms the ITLCO algorithm on these nine test functions. The Wilcoxon rank sum test yielded a negative result for the ITLCO1 algorithm on the nine test functions, suggesting that its performance is significantly inferior to that of the ITLCO algorithm on these specific functions. Conversely, the Wilcoxon rank sum test yielded a positive result for the remaining 14 test functions, indicating that their performance is comparable. The TLCO2 algorithm demonstrates superior performance compared to the ITLCO algorithm on a single test function, exhibits similar performance to the ITLCO algorithm on 12 test functions, and demonstrates significant superiority over the ITLCO algorithm on 15 test functions. Conversely, the ITLCO3 algorithm demonstrates significantly better performance than the ITLCO algorithm on two test functions, demonstrates performance comparable to that of the ITLCO algorithm across thirteen test functions, and performs inferiorly to the ITLCO algorithm on thirteen test functions. Furthermore, it is observed that the rank average of the improvement algorithms, after eliminating the corresponding improvement strategy in ITLCO, surpasses that of the ITLCO algorithm. This suggests that the three enhancement strategies proposed in this study have a significant impact on the overall performance of the ITLCO algorithm. Specifically, the enhancement strategies described in “Strategies for the Generation of New Workers” and “Strategies for the Generation of New Soldiers” have a more significant effect on the performance of the ITLCO algorithm, whereas the enhancement strategies outlined in “Novel Mechanism for Replacement Renewal” show only a minor difference in their effect on the performance of the ITLCO algorithm. The impact of the enhancement methods outlined in “Novel Mechanism for Replacement Renewal” on the performance of the ITLCO algorithm is minimal.

This research demonstrates the correctness of all the three recommended enhancements.

Comparative analysis of the ITLCO algorithm in relation to other algorithms

To conduct a thorough evaluation of the ITLCO’s performance, this section compares it with the TLCO algorithm and the four more excellent evolutionary algorithms to date on the CEC2013 test set, CEC2019 test set and CEC2020 test set, including MSMA (Deng & Liu, 2023), IECO (Wang & Liu, 2023), IAOA (Wang, Chen & Ku, 2023), and ESO (Yao et al., 2023). Among them, the dimensions of each test function in CEC2013 test set are 30, and each test function in CEC2020 test set is 10, while the dimensions of F1-F3 in CEC2019 test set are 9, 16, 18, and 10 in F4–F10.

To maintain fairness in the comparison, all algorithms have a population size of N = 50, and On CEC2013 test set and CEC2020 test set, the termination condition is MaxFEs = 100,000, and on CEC2019 test set, the termination condition is MaxIts = 500. Table 5 displays the additional parameter settings for each method, ensuring that the parameter values obtained from each comparison algorithm align with the original text.

Table 5 Initial parameter settings for each algorithm.

Algorithm	Initial parameters	
ITLCO	p = 5	
TLCO	None	
MSMA	z=0.03, E=100, epoch= 0, j=0	
IECO	numberC=5; Swarmsize = 10; Nmat = 3; limit = 10	
IAOA	C2=6; C3=2; C4=0.5	
ESO	Threshold=0.25; Thresold2=0.6; T=900; vec_flag=[1, −1]	

Comparison of ITLCO algorithm with other algorithms regarding convergence accuracy

To assess the convergence accuracy of the ITLCO algorithm in comparison to other algorithms, it was evaluated using the CEC2013 test set, CEC2019 test set and CEC2020 test set with a depth of 30. Tables 6–8 provides the average and variability from 30 individual experiments for each algorithm applied to the 30-dimensional CEC2013 dataset. The function that demonstrates the highest optimization on the same test function is highlighted in bold. In order to conduct a more comprehensive analysis of the distinctions between the ITLCO algorithm and the other algorithms, The results of Wilcoxon rank sum test with a significance level of 5% and Friedman test between ITLCO algorithm and each improved algorithm are shown in Tables 9–11.

Table 6 Data results of ITLCO algorithm and other algorithms on the 30-dimensional CEC2013 test set.

	ITLCO	TLCO	MSMA	IECO	IAOA	ESO	
F1	0.00E+00 ± 0.00E+00	6.65E−01 ± 5.44E−01	9.23E+00 ± 3.37E+01	8.41E−30 ± 3.71E−29	0.00E+00 ± 0.00E+00	8.77E+00 ± 4.71E+00	
F2	7.73E+05 ± 4.94E+05	1.97E+07 ± 1.05E+07	3.53E+07 ± 1.50E+07	1.89E+06 ± 1.56E+06	1.27E+06 ± 1.24E+06	1.46E+07 ± 3.81E+06	
F3	0.00E+00 ± 0.00E+00	0.00E+00 ± 0.00E+00	0.00E+00 ± 0.00E+00	0.00E+00 ± 0.00E+00	0.00E+00 ± 0.00E+00	0.00E+00 ± 0.00E+00	
F4	9.15E+03 ± 2.76E+03	1.88E+04 ± 3.50E+03	5.09E+04 ± 5.17E+03	2.52E+04 ± 9.67E+03	8.04E+03 ± 9.17E+03	3.92E+04 ± 7.77E+03	
F5	0.00E+00 ± 0.00E+00	1.45E+00 ± 9.70E−01	1.12E+01 ± 1.10E+01	2.05E−30 ± 1.02E−29	0.00E+00 ± 0.00E+00	1.73E+00 ± 1.63E+00	
F6	1.36E+01 ± 1.63E+01	8.73E+01 ± 3.05E+01	9.64E+01 ± 2.74E+01	3.92E+01 ± 3.25E+01	2.76E+01 ± 2.38E+01	8.57E+01 ± 3.40E+01	
F7	0.00E+00 ± 0.00E+00	0.00E+00 ± 0.00E+00	9.40E+00 ± 2.72E+01	0.00E+00 ± 0.00E+00	0.00E+00 ± 0.00E+00	3.30E+00 ± 1.51E+01	
F8	2.10E+01 ± 6.24E−02	2.10E+01 ± 5.96E−02	2.10E+01 ± 7.16E−02	2.12E+01 ± 6.21E−02	2.10E+01 ± 6.53E−02	2.10E+01 ± 5.25E−02	
F9	0.00E+00 ± 0.00E+00	2.17E+00 ± 6.59E+00	4.37E+00 ± 9.88E+00	0.00E+00 ± 0.00E+00	0.00E+00 0.00E+00	7.43E+00 ± 9.89E+00	
F10	7.55E−02 ± 4.28E−02	2.62E+01 ± 1.56E+01	9.15E+01 ± 8.30E+01	2.77E−01 ± 1.08E-01	8.05E−02 ± 7.42E−02	9.63E+00 ± 3.04E+00	
F11	0.00E+00 ± 0.00E+00	0.00E+00 ± 0.00E+00	0.00E+00 ± 0.00E+00	0.00E+00 ± 0.00E+00	0.00E+00 ± 0.00E+00	0.00E+00 ± 0.00E+00	
F12	0.00E+00 ± 0.00E+00	1.50E+02 ± 4.20E+01	1.20E+02 ± 7.79E+01	0.00E+00 ± 0.00E+00	0.00E+00 ± 0.00E+00	6.77E+01 ± 6.53E+01	
F13	0.00E+00 ± 0.00E+00	1.51E+02 ± 6.20E+01	1.34E+02 ± 9.06E+01	0.00E+00 ± 0.00E+00	0.00E+00 ± 0.00E+00	8.16E+01 ± 7.82E+01	
F14	9.09E+02 ± 2.26E+02	6.48E+02 ± 2.19E+02	1.56E+03 ± 4.33E+02	1.32E+03 ± 3.49E+02	2.17E+03 ± 4.13E+02	1.81E+03 ± 3.83E+02	
F15	4.27E+03 ± 7.27E+02	4.21E+03 ± 6.70E+02	4.57E+03 ± 8.93E+02	2.72E+03 ± 5.18E+02	6.92E+03 ± 4.94E+02	6.58E+03 ± 3.90E+02	
F16	1.49E+00 ± 4.01E−01	1.38E+00 ± 4.14E−01	2.38E+00 ± 5.33E−01	4.27E+00 ± 6.57E−01	2.80E+00 ± 3.61E−01	2.22E+00 ± 2.86E−01	
F17	3.81E+01 ± 8.99E+00	4.92E+01 ± 1.27E+01	2.89E+02 ± 1.47E+02	4.07E+01 ± 1.52E+01	4.36E+01 ± 2.06E+01	4.09E+02 ± 6.90E+01	
F18	9.91E+01 ± 3.04E+01	2.18E+02 ± 4.82E+01	3.88E+02 ± 1.25E+02	2.15E+02 ± 3.47E+01	1.81E+02 ± 2.66E+01	4.09E+02 ± 7.04E+01	
F19	5.63E+00 ± 2.21E+00	1.63E+01 ± 5.83E+00	3.01E+01 ± 9.79E+00	3.34E+00 ± 9.19E−01	1.18E+01 ± 2.88E+00	2.46E+01 ± 3.82E+00	
F20	0.00E+00 ± 0.00E+00	1.05E+01 ± 4.21E+00	1.50E+01 ± 3.29E−07	0.00E+00 ± 0.00E+00	0.00E+00 ± 0.00E+00	7.65E+00 ± 4.48E+00	
F21	4.00E+02 ± 0.00E+00	4.00E+02 ± 4.23E−01	4.00E+02 ± 1.37E−01	4.00E+02 ± 0.00E+00	3.93E+02 ± 3.59E+01	4.02E+02 ± 6.91E−01	
F22	8.71E+02 ± 3.07E+02	5.83E+02 ± 2.04E+02	1.60E+03 ± 4.00E+02	1.35E+03 ± 4.47E+02	1.93E+03 ± 5.04E+02	2.02E+03 ± 4.46E+02	
F23	4.85E+03 ± 6.79E+02	5.22E+03 ± 7.45E+02	5.12E+03 ± 7.73E+02	3.02E+03 ± 6.47E+02	6.65E+03 ± 5.54E+02	7.13E+03 ± 4.52E+02	
F24	2.00E+02 ± 4.14E-02	2.02E+02 ± 8.67E+00	2.49E+02 ± 3.96E+01	2.01E+02 ± 2.92E+00	2.29E+02 ± 3.38E+01	2.11E+02 ± 2.25E+01	
F25	2.32E+02 ± 2.18E+01	2.98E+02 ± 3.38E+01	2.97E+02 ± 8.18E+00	3.00E+02 ± 1.12E+01	2.82E+02 ± 1.94E+01	2.70E+02 ± 2.54E+01	
F26	2.93E+02 ± 2.49E+01	2.90E+02 ± 4.44E+01	3.57E+02 ± 4.02E+01	2.90E+02 ± 4.44E+01	3.23E+02 ± 2.78E+01	3.27E+02 ± 3.11E+01	
F27	3.15E+02 ± 9.32E-01	3.76E+02 ± 1.84E+02	9.14E+02 ± 3.25E+02	3.50E+02 ± 1.43E+02	8.68E+02 ± 2.38E+02	7.45E+02 ± 2.60E+02	
F28	8.75E+02 ± 2.09E+01	1.08E+03 ± 6.03E+01	1.22E+03 ± 4.22E+02	9.47E+02 ± 4.58E+02	1.05E+03 ± 5.09E+02	3.00E+03 ± 5.46E+02	
Note:

The function that demonstrates the highest optimization on the same test function is highlighted in bold.

Table 7 Results of ITLCO algorithm and other algorithms on the CEC2019 test set.

Function	ITLCO	TLCO	MSMA	IECO	IAOA	ESO	
F1	9.21E+04 ± 7.34E+04	1.00E+00 ± 2.12E-10	1.00E+00 ± 0.00E+00	5.56E+08 ± 2.15E+08	4.86E+01 ± 1.10E+02	1.00E+00 ± 0.00E+00	
F2	2.47E+02 ± 1.13E+02	4.50E+00 ± 3.31E−01	4.74E+00 ± 3.24E−01	1.96E+04 ± 5.29E+03	5.07E+00 ± 8.54E−01	4.56E+00 ± 2.83E−01	
F3	1.51E+00 ± 5.91E−01	2.40E+00 ± 8.86E−01	2.55E+00 ± 1.95E+00	1.22E+01 ± 4.14E−01	4.08E+00 ± 1.58E+00	4.16E+00 ± 6.53E−01	
F4	1.30E+01 ± 5.56E+00	1.93E+01 ± 7.56E+00	1.86E+01 ± 6.69E+00	1.10E+02 ± 1.91E+01	5.40E+00 ± 1.75E+00	1.60E+01 ± 6.28E+00	
F5	1.11E+00 ± 6.88E−02	1.32E+00 ± 1.43E−01	1.23E+00 ± 8.21E−02	8.71E+01 ± 1.84E+01	1.02E+00 ± 1.32E−02	1.64E+00 ± 1.38E−01	
F6	1.04E+00 ± 1.89E−01	3.05E+00 ± 1.25E+00	3.59E+00 ± 1.76E+00	1.31E+01 ± 9.68E−01	2.61E+00 ± 1.51E+00	2.78E+00 ± 7.54E−01	
F7	6.24E+02 ± 2.93E+02	6.60E+02 ± 2.54E+02	7.52E+02 ± 2.43E+02	2.54E+03 ± 2.21E+02	5.37E+02 ± 2.53E+02	7.40E+02 ± 3.35E+02	
F8	3.48E+00 ± 3.83E−01	3.81E+00 ± 4.63E−01	3.60E+00 ± 4.14E−01	5.39E+00 ± 1.46E−01	3.80E+00 ± 4.01E−01	4.00E+00 ± 3.53E−01	
F9	1.11E+00 ± 4.84E−02	1.28E+00 ± 1.06E−01	1.23E+00 ± 6.88E−02	3.99E+00 ± 6.50E−01	1.15E+00 ± 3.21E−02	1.20E+00 ± 5.58E−02	
F10	1.98E+01 ± 4.64E+00	2.04E+01 ± 3.16E+00	2.06E+01 ± 3.49E+00	2.20E+01 ± 1.20E-01	2.13E+01 ± 8.74E-02	2.05E+01 ± 3.20E+00	

Table 8 Results of ITLCO algorithm and other algorithms on the CEC2020 test set.

Function	ITLCO	TLCO	MSMA	IECO	IAOA	ESO	
F1	7.66E+02 ± 1.09E+03	1.73E+03 ± 1.87E+03	6.04E+03 ± 4.42E+03	8.45E+02 ± 1.05E+03	3.19E+05 ± 2.34E+00	5.72E+05 ± 3.85E+05	
F2	2.85E+02 ± 2.25E+02	3.13E+02 ± 1.86E+02	6.21E+02 ± 1.67E+02	3.62E+02 ± 2.12E+02	6.93E+02 ± 2.18E+02	2.73E+02 ± 2.97E+02	
F3	1.77E+01 ± 4.47E+00	2.68E+01 ± 7.32E+00	4.50E+01 ± 1.25E+01	2.07E+01 ± 4.20E+00	1.59E+01 ± 1.42E+00	3.92E+01 ± 7.05E+00	
F4	7.82E−01 ± 1.22E+03	1.54E+00 ± 6.17E−01	2.18E+00 ± 1.12E+00	1.04E+00 ± 5.39E−01	9.83E−01 ± 2.19E−01	2.13E+00 ± 6.01E−01	
F5	1.22E+03 ± 2.12E+01	4.23E+03 ± 4.35E+03	2.09E+04 ± 6.97E+04	1.92E+03 ± 2.00E+03	1.05E+04 ± 3.11E+03	9.70E+03 ± 1.88E+04	
F6	5.81E+00 ± 7.72E+01	7.30E+01 ± 8.40E+01	1.64E+02 ± 9.37E+01	9.32E+01 ± 9.12E+01	7.36E+01 ± 6.98E+01	9.79E+01 ± 5.32E+01	
F7	8.74E+02 ± 6.85E+02	2.42E+03 ± 2.04E+03	5.45E+03 ± 6.03E+03	7.96E+02 ± 6.08E+02	2.13E+05 ± 6.35E+05	3.22E+03 ± 2.31E+03	
F8	1.08E+02 ± 2.20E+03	1.10E+02 ± 2.50E-06	1.10E+02 ± 5.35E-03	1.10E+02 ± 5.68E-14	4.67E+01 ± 2.97E+01	1.10E+02 ± 1.26E-01	
F9	3.01E+02 ± 7.27E+01	2.95E+02 ± 9.96E+01	3.62E+02 ± 1.03E+01	3.25E+02 ± 6.05E+01	3.34E+02 ± 3.80E+01	3.44E+02 ± 8.02E+00	
F10	4.21E+02 ± 2.22E+01	4.27E+02 ± 2.32E+01	4.39E+02 ± 3.79E+01	4.30E+02 ± 2.11E+01	4.30E+02 ± 2.28E+01	4.30E+02 ± 2.13E+01	

Table 9 Wilcoxon rank sum test results for ITLCO and other algorithms on the CEC2013 test set.

Function	p-value (vs.ITLCO)	
TLCO	MSMA	IECO	IAOA	ESO	
F1	0.000(−)	0.000(−)	0.161(=)	1.000(=)	0.000(−)	
F2	0.000(−)	0.000(−)	0.000(−)	0.066(=)	0.000(−)	
F3	1.000(=)	1.000(=)	1.000(=)	1.000(=)	1.000(=)	
F4	0.000(−)	0.000(−)	0.000(−)	0.004(+)	0.000(−)	
F5	0.000(−)	0.000(−)	0.000(−)	1.000(=)	0.000(−)	
F6	0.000(−)	0.000(−)	0.003(−)	0.000(−)	0.000(−)	
F7	1.000(=)	0.011(−)	1.000(=)	1.000(=)	0.161(=)	
F8	0.795(=)	0.001(−)	0.000(−)	0.002(−)	0.938(=)	
F9	0.082(=)	0.022(−)	1.000(=)	1.000(=)	0.000(−)	
F10	0.000(−)	0.000(−)	0.000(+)	0.779(=)	0.000(−)	
F11	1.000(=)	1.000(=)	1.000(=)	1.000(=)	1.000(=)	
F12	0.000(−)	0.000(−)	1.000(=)	1.000(=)	0.000(−)	
F13	0.000(−)	0.000(−)	1.000(=)	1.000(=)	0.000(−)	
F14	0.000(+)	0.000(−)	0.000(−)	0.000(−)	0.000(−)	
F15	0.564(=)	0.178(=)	0.000(−)	0.000(−)	0.000(−)	
F16	0.412(=)	0.000(−)	0.000(−)	0.000(−)	0.000(−)	
F17	0.000(−)	0.000(−)	0.790(=)	0.756(=)	0.000(−)	
F18	0.000(−)	0.000(−)	0.000(−)	0.000(−)	0.000(−)	
F19	0.000(−)	0.000(−)	0.000(+)	0.000(−)	0.000(−)	
F20	0.000(−)	0.000(−)	1.000(=)	1.000(=)	0.000(−)	
F21	0.005(−)	0.001(−)	1.000(=)	0.334(=)	0.000(−)	
F22	0.000(−)	0.000(−)	0.000(−)	0.000(−)	0.000(−)	
F23	0.117(=)	0.234(=)	0.000(+)	0.000(−)	0.000(−)	
F24	0.000(−)	0.000(−)	0.006(−)	0.000(−)	0.000(−)	
F25	0.000(−)	0.000(−)	0.000(+)	0.000(−)	0.000(−)	
F26	0.782(=)	0.000(−)	0.347(=)	0.000(−)	0.000(−)	
F27	0.000(−)	0.000(−)	0.023(−)	0.000(−)	0.000(−)	
F28	0.000(−)	0.000(−)	0.000(−)	0.000(−)	0.000(−)	
+/=/−	1/9/18	0/4/24	5/11/12	1/13/14	0/4/24	

Table 10 Wilcoxon rank sum test results of ITLCO and other algorithms on the CEC2019 test set.

Function	p-value (vs.ITLCO)	
TLCO	MSMA	IECO	IAOA	ESO	
F1	0.000(+)	0.000(+)	0.000(−)	0.000(+)	0.000(+)	
F2	0.000(+)	0.000(+)	0.000(−)	0.000(+)	0.000(+)	
F3	0.000(−)	0.000(−)	0.000(−)	0.000(−)	0.000(−)	
F4	0.001(−)	0.000(−)	0.000(−)	0.000(+)	0.047(−)	
F5	0.000(−)	0.000(−)	0.000(−)	0.000(+)	0.000(−)	
F6	0.000(−)	0.000(−)	0.000(−)	0.000(−)	0.000(−)	
F7	0.046(=)	0.069(=)	0.000(−)	0.054(=)	0.252(=)	
F8	0.005(−)	0.433(=)	0.000(−)	0.002(−)	0.000(−)	
F9	0.000(−)	0.000(−)	0.000(−)	0.013(−)	0.000(−)	
F10	0.190(=)	0.000(−)	0.000(−)	0.009(−)	0.000(−)	
+/=/−	2/2/6	2/2/6	0/0/10	4/1/5	2/1/7	

Table 11 Wilcoxon rank sum test results of ITLCO and other algorithms on the CEC2020 test set.

Function	p-value (vs.ITLCO)	
TLCO	MSMA	IECO	IAOA	ESO	
F1	0.003(−)	0.000(−)	0.877(=)	0.000(+)	0.000(−)	
F2	0.473(=)	0.000(−)	0.050(=)	0.000(−)	0.412(=)	
F3	0.000(−)	0.000(−)	0.014(−)	0.000(+)	0.000(−)	
F4	0.000(−)	0.000(−)	0.099(=)	0.663(=)	0.000(−)	
F5	0.001(−)	0.000(−)	0.274(=)	0.712(=)	0.000(−)	
F6	0.000(−)	0.000(−)	0.000(−)	0.000(−)	0.000(−)	
F7	0.001(−)	0.000(−)	0.739(=)	0.002(−)	0.000(−)	
F8	0.334(=)	0.334(=)	0.334(=)	0.289(=)	0.168(=)	
F9	0.031(+)	0.000(−)	0.428(=)	0.335(=)	0.026(−)	
F10	0.007(−)	0.001(−)	0.145(=)	0.009(−)	0.001(−)	
+/=/−	1/2/7	0/1/9	0/8/2	2/4/4	0/2/8	

On the CEC2013 test set, Table 6 shows that ITLCO obtains global optimal values on nine test functions, including F1, F3, F5, F7, F9, F11, F12, F13, and F20. As shown in Tables 6 and 9, for the unimodal function F1–F5, TLCO, MSMA and ESO are only equal to ITLCO on F3 and worse than ITLCO on the four other test functions. IECO is equal to ITLCO only on F1 and F3 and performs worse than ITLCO on the three other test functions. IAOA performs better than ITLCO only on F4 and is equal to ITLCO on the four other test functions. Therefore, the performance of ITLCO on unimodal functions is only worse than IAOA but better than the other comparison algorithms. For multi-modal functions F6–F20, TLCO and IECO are superior to ITLCO in only one test function; there are six and seven equal to ITLCO, but their performance is worse than ITLCO in eight and seven test functions, respectively. MSMA, IAOA, and ESO are equal to ITLCO in two, eight and three test functions, respectively, but perform worse than ITLCO in 13, seven, and 12 test functions, respectively. Therefore, ITLCO is superior to other comparison algorithms on multi-modal functions. For compound functions F21–F28, IECO performs better than ITLCO in only two test functions, is equal to ITLCO in two test functions and performs worse than ITLCO in four test functions. MSMA, IAOA, and ESO are equal to ITLCO in only one, one and zero test functions, respectively, while their performance is worse than ITLCO in 7, 7, and 8 test functions, respectively. Therefore, ITLCO is obviously superior to other comparison algorithms on composite functions. In conclusion, for the 30-dimensional CEC2013 test set, the ITLCO proposed in this article has obvious advantages in convergence accuracy compared with other representative methods.

On the CEC2019 test set, Tables 7 and 10 shows that ITLCO only performs significantly worse on F1 and F2 and performs similarly on F7 and F10 for multi-modal functions compared with TLCO. ITLCO achieves better average values on all six test functions, and its performance is significantly better. Compared with MSMA, ITLCO has poor performance on F1 and F2, similar performance on F7 and F10, and significantly better performance on six test functions. Compared with IECO, ITLCO has better meaning value and better performance in all 10 test functions. Compared with IAOA, ITLCO has poor performance on four test functions, similar performance on F7, and significantly better performance on five test functions. Compared with ESO, ITLCO has poor performance on F1 and F2, similar performance on F7 only, and significantly better performance on seven test functions. Therefore, ITLCO is superior to other comparison algorithms on multi-modal functions. Based on the rank sum check results, ITLCO has certain advantages on the CEC2019 test set compared with other comparison algorithms.

To further test the performance of the ITLCO algorithm, the ITLCO algorithm and four other algorithms were tested separately on the CEC2020 test set. As shown in Tables 8 and 11 on the 10-dimensional CEC2020 test set, for unimodal function F1, ITLCO’s performance is only worse than IAOA, the same as IECO, but better than the other comparison algorithms. For multi-modal functions F2–F4, TLCO, MSMA, IECO, and ESO have the same performance as ITLCO in one, zero, two and one, respectively, while the performance of two, three, one and two test functions is worse than ITLCO. Compared with ITLCO, IAOA has 1 better, equal, and worse test function. Therefore, ITLCO is equal to IAOA and better than the other algorithms on multi-modal functions. For the mixed function F5–F7, TLCO, MSMA, and ESO perform worse than ITLCO in three cases respectively. The performance of IECO and IAOA is equal to that of ITLCO in two and one, respectively, while the performance of IECO and IAOA in one and two test functions is worse than that of ITLCO. Therefore, ITLCO is obviously superior to other algorithms on mixed functions. For the mixed function F8-F10, MSMA, IECO, IAOA, and ESO have the same performance as ITLCO in one, three, two, and two test functions, respectively, and the performance on two, zero, one, and one test functions is worse than ITLCO. Compared with ITLCO, TLCO has 1 better, equal, and worse test function. Therefore, ITLCO is equal to TLCO and superior to other algorithms in terms of composite functions. In conclusion, ITLCO has certain advantages on the CEC2020 test set compared with the other algorithms.

Figure 1 shows the rank means of the Friedman test statistic, where in the CEC2013 test set, CEC2019 test set, and CEC2020 test set, ITLCO has the smallest rank mean, thus demonstrating that ITLCO has the best overall performance among the six optimization algorithms. This indicates that, compared with other algorithms, the ITLCO proposed in this article has certain advantages in convergence accuracy.

Figure 1 Friedman tests of ITLCO and other algorithms on different test sets.

Comparison of ITLCO algorithm with other algorithms regarding convergence rate

In order to compare the complexity and convergence speed of the ITLCO algorithm with the other algorithms, Table 12 gives the average running time of each algorithm for 30 independent runs of 2,000 generations each on the 30-dimensional CEC2013 test set, and Table 13 gives the average running time of each algorithm when it reaches the same convergence accuracy. The other parameter settings are exactly the same as in “Comparison of ITLCO Algorithm with Other Algorithms Regarding Convergence Accuracy”.

Table 12 The running time of each algorithm on the CEC2013 test set (/s).

Function	ITLCO	TLCO	MSMA	IECO	IAOA	ESO	
F1	5.8	5.1	3.5	8.9	1.3	3.9	
F2	6.3	6.2	12.6	11.1	2.5	13.0	
F3	5.9	5.7	7.9	10.2	1.7	9.6	
F4	6.0	5.8	8.9	10.1	2.0	9.6	
F5	6.4	5.4	6.9	9.3	1.5	7.1	
F6	12.1	5.4	5.4	9.2	1.5	8.9	
F7	7.7	7.6	25.0	13.9	3.7	38.1	
F8	7.2	6.9	18.9	12.3	3.5	29.7	
F9	24.4	24.0	200.7	52.7	29.6	220.8	
F10	6.5	6.2	12.2	10.7	2.5	24.2	
F11	6.4	6.2	12.5	11.0	2.4	11.8	
F12	7.3	7.2	20.5	13.0	3.5	38.2	
F13	8.3	8.0	27.5	14.4	4.4	45.3	
F14	6.0	5.9	10.4	10.2	2.2	9.9	
F15	6.6	6.4	13.5	11.0	2.7	19.9	
F16	18.5	18.3	140.1	39.1	21.4	152.1	
F17	5.9	5.8	9.6	9.9	2.1	9.1	
F18	6.3	6.2	12.8	10.8	2.5	13.9	
F19	5.7	5.6	6.8	9.5	1.7	6.9	
F20	6.2	6.1	8.2	11.0	2.0	14.6	
F21	10.2	9.5	39.7	17.6	6.5	36.9	
F22	9.5	9.2	39.9	17.1	6.5	31.4	
F23	10.0	9.8	45.3	18.2	7.4	57.0	
F24	29.2	28.9	245.4	65.2	36.9	287.0	
F25	29.1	28.9	233.0	65.9	37.0	290.2	
F26	31.4	31.1	249.4	70.0	40.8	320.7	
F27	30.7	30.4	261.3	69.2	39.8	313.8	
F28	13.5	12.5	70.3	25.0	11.1	87.0	

Table 13 Comparison of time for each algorithm when run to the same convergence accuracy on the CEC2013 test set (/s).

Function	Preset accuracy	ITLCO	TLCO	MSMA	IECO	IAOA	ESO	
F1	1.00E+01	1.07	1.84	0.64	1.42	1.09	6.26	
F2	1.00E+08	0.44	0.41	0.51	0.44	1.27	18.65	
F3	1.00E-05	0.33	0.44	0.06	0.32	0.49	13.00	
F4	1.00E+05	0.11	0.42	0.03	0.11	0.64	13.65	
F5	1.00E+02	0.72	1.04	0.49	1.06	0.78	10.61	
F6	1.00E+02	1.28	2.20	0.62	3.01	1.20	8.73	
F7	1.00E+02	0.26	0.19	0.07	0.18	0.36	33.92	
F8	1.00E+02	0.01	0.02	0.00	0.02	0.02	27.04	
F9	1.00E+01	1.06	4.63	21.05	1.92	1.49	273.66	
F10	1.00E+02	1.08	2.02	1.22	1.18	0.16	18.08	
F11	1.00E-05	0.16	0.14	0.00	0.28	0.02	18.55	
F12	2.00E+02	0.55	1.08	0.11	0.14	0.91	30.61	
F13	2.00E+02	0.81	0.90	0.13	0.24	0.37	39.13	
F14	3.00E+03	1.85	1.79	5.64	1.24	1.51	15.84	
F15	7.00E+03	1.64	1.47	5.41	2.91	2.07	19.53	
F16	5.00E+00	0.11	0.13	0.18	0.36	0.11	191.48	
F17	1.00E+03	0.09	0.12	0.08	0.06	0.02	14.35	
F18	1.00E+03	0.11	0.11	0.09	0.05	0.03	18.65	
F19	1.00E+02	0.42	0.51	0.32	0.13	0.06	10.93	
F20	1.00E+01	1.83	3.39	6.95	6.77	0.28	18.65	
F21	1.00E+03	0.37	0.43	0.56	0.28	0.13	61.45	
F22	3.00E+03	2.82	1.61	20.48	2.55	4.23	61.49	
F23	8.00E+03	2.00	0.83	17.03	2.86	3.34	65.32	
F24	3.00E+02	0.40	0.47	0.48	0.17	0.25	343.02	
F25	4.00E+02	0.06	0.14	0.04	0.09	0.31	344.38	
F26	4.00E+02	0.10	0.26	5.27	0.13	0.29	371.96	
F27	1.00E+03	0.47	1.45	1.02	0.23	0.37	354.80	
F28	4.00E+03	0.08	0.18	0.24	0.07	0.12	95.74	
		20.23	28.23	88.73	28.25	21.94	2,499.46	

As shown in Table 12, ITLCO has a slightly larger time consumption than the original algorithm TLCO mainly because ITLCO proposes a new worker generation strategy and a new soldier generation strategy and introduces a new replacement and update mechanism into the algorithm, resulting in higher complexity than the original algorithm TLCO. From Table 13, it can be seen that the ITLCO algorithm runs in less time than the other algorithms when the same convergence accuracy is achieved. In summary, compared with other comparison algorithms, the proposed ITLCO algorithm has a shorter running time and does not consume more running time to improve the convergence performance of the algorithm, and the running time is on the order of magnitude with TLCO.

In addition, for a more intuitive comparison of the differences in convergence rates between algorithms, according to Fig. 2, ITLCO demonstrates the ability to attain the global optimum on the F1, F3, and F5 test functions for single-mode functions F1–F5. However, it exhibits a slightly slower convergence speed compared to IECO and ESO for F3. Additionally, ITLCO demonstrates superior convergence accuracy, and the fastest convergence speed relative to the other five algorithms for F2 and F4. For multimodal functions F6–F20, ITLCO is able to achieve the global optimum on the F7, F9, F11–13, and F20, ITLCO exhibits superior convergence accuracy and the fastest convergence speed compared to the other algorithms on F6, F8, F10, F14–18 and F20; in the pre-evolutionary stage, ITLCO converges somewhat more slowly than the other algorithms, and when it arrives at the post-evolutionary stage, ITLCO is able to continue to search for better solution, whereas the other algorithms experience a decline in their search performance; ITLCO on F19 is second only to IECO in terms of convergence accuracy and convergence speed. For compound function F21–F28, the convergence accuracy of ITLCO on F23 is better than the other five algorithms, and the convergence speed is faster. Additionally, on F22 and F24, ITLCO exhibits higher convergence accuracy and faster convergence speed. The convergence speed and convergence accuracy of ITLCO on F21 and F27 exhibit similarities to those of IECO. On F25, the convergence speeds of ITLCO and ESO are comparable, with ITLCO demonstrating higher convergence accuracy. On F26 and F28, the convergence speed and convergence accuracy of ITLCO are comparable to that of IECO. In conclusion, ITLCO has a superior convergence speed in comparison to the remaining five methods.

Figure 2 The convergence curves of each algorithm on the 28 test functions of the CEC2013 test set for a dimension of 30.

Conclusions

This study introduces an ITLCO to enhance the convergence performance and capacity to overcome the local optima of the TLCO. A novel approach is developed for the worker generation, wherein various updating methods are chosen based on the individual goodness of fitness. This strategy aims to enhance convergence speed while preserving population diversity. Additionally, it aims to improve communication between the worker population and the termite population, thereby increasing the likelihood of the algorithm successfully escaping local optima. Furthermore, a novel approach to soldier generation is suggested, which incorporates random vectors that adhere strictly to the principles of evolution and differ from one another in every dimension. These vectors serve as the step factor to enhance the algorithm’s convergence speed. Additionally, a novel mechanism for replacing updates is introduced, which incorporates an adaptive crossover factor to strike a balance between the algorithm’s convergence and the diversity of the population. The simulation results obtained from conducting multiple experiments on the CEC2013, CEC2019and CEC2020 test set demonstrate that the proposed improvement strategy significantly enhances the overall performance of TLCO. Furthermore, ITLCO exhibits notable advantages in terms of convergence speed and solution accuracy when compared with various other algorithms. However, for high-dimensional and complex optimization problems, the algorithm complexity and running time will increase greatly, and the algorithm performance will decrease. How to reduce the complexity of the algorithm and improve the performance of the algorithm in complex optimization problems remain unclear, and their effect should be improved for practical applications. Maximizing the benefit will be the focus of future work.

Supplemental Information

Supplemental Information 1 ITLCO algorithm and iteration graph.

Supplemental Information 2 Comparative algorithms and improved effectiveness analysis.

Supplemental Information 3 Original data from CEC2013: Function evaluation criteria and details of CEC2013 test set.

The original source of CEC2013 is divided into two parts: the first part is the functional evaluation criteria and details of the CEC2013 test set, which can be found in Appendix CEC2013. The second part is the relevant data files needed to test the CEC2013 test set, which can be found in the code files (e.g., input_data, cec13, etc.,).

Supplemental Information 4 CEC2019 related benchmark function.

Supplemental Information 5 CEC2020 Related benchmark functions.

Additional Information and Declarations

Competing Interests

The authors declare that they have no competing interests.

Author Contributions

Yanjiao Wang conceived and designed the experiments, performed the experiments, analyzed the data, performed the computation work, prepared figures and/or tables, authored or reviewed drafts of the article, and approved the final draft.

Mengjiao Wei conceived and designed the experiments, performed the experiments, analyzed the data, performed the computation work, prepared figures and/or tables, authored or reviewed drafts of the article, and approved the final draft.

Data Availability

The following information was supplied regarding data availability:

The original data is available in the Supplemental Files.

The code is available in the Supplemental Files and at Zenodo: Weimmmmm. (2024). Weimmmmm/ITLCO-Code: first release of my awesome software (V1.0.0). Zenodo. https://doi.org/10.5281/zenodo.13922727.

The data is available at GitHub and Zenodo:

- https://github.com/Weimmmmm/ITLCO-Date

- Weimmmmm. (2025). Weimmmmm/ITLCO-Date: first (v0.1.0). Zenodo. https://doi.org/10.5281/zenodo.14642295

- https://github.com/Weimmmmm/TLCO_CEC2013

- Weimmmmm. (2025). Weimmmmm/TLCO_CEC2013: first1 (v.1.0.0). Zenodo. https://doi.org/10.5281/zenodo.14642306.

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
