# Peer review of "An improved termite life cycle optimizer algorithm for global function optimization"

_PeerJ Computer Science, doi:10.7717/peerj-cs.2671_

## Round 0.1 · original submission · Major Revisions

Dear Authors,

Reviewers' comments on your submission have now been received. Your article has not been recommended for publication in its current form. However, we do encourage you to address the concerns and criticisms of the reviewers and resubmit your article once you have updated it accordingly. Before submitting the revised paper, following should also be addressed:

1. It is not clear what this improved method teaches us on behavior of termites.
2. It is not possible to ascertain the precise nature of the differences between this algorithm and existing algorithms, or indeed to determine whether it is distinct from the numerous other metaphor-inspired methods that are currently in use.
3. The reason to assume that the behavior of termit would be a good inspiration to solve engineering problems is not clear. Similarly, the motivation and reason of using Termit Life Cycle among many other metaheuristic algorithms for function optimization should be mentioned.
4. Pay special attention to the usage of abbreviations. Spell out the full term at its first mention, indicate its abbreviation in parenthesis and use the abbreviation from then on.
5. Equations should be used with correct equation number. Many of the equations are part of the related sentences. Attention is needed for correct sentence formation. Equation numbering should be corrected.
6. All of the values for the parameters of all algorithms should be given.

Best wishes,

Reviewer 1 ·

Basic reporting

no comment

Experimental design

no comment

Validity of the findings

the comparison for CEC2019, C2020 must be presented

Additional comments

1. Does the ITLCO with new strategies impact the computational cost of the ITLCO algorithm compared to other algorithms?
2. What is the basis for asserting that strategies for the Generation of New Workers achieve a harmonious equilibrium between algorithm convergence and population diversity? Please clarify in a visual format.
3. Does the ITLCO algorithm still use a 7:3 ratio of worker to soldier individuals? Has this ratio been optimized?
4. Based on the results in Table 5, there are certain functions like F16, F22, etc., where ITLCO is less effective than TLCO. Could the reason be that the improvements cause the step size after each update to become larger, making it harder to accurately reach the extremum in functions with high sharpness? Please show more information about the different of step size between TLCO and ITLCO.
5. What would happen if Xg were equal to Xbest or XPbest? What is the reason to use a random coefficient with a decision threshold of 0.5 instead of basing it on other characteristics of the dataset?
Additional descriptions of equations (10) and (11) are needed to clarify how the proposed equations differ from the original TLCO.
6. Adding options for ITLCO, such as equations (10) and (11), will increase the computation time compared to the original algorithm. I strongly recommend comparing the execution times of both algorithms.
7. Additional benchmarks, such as CEC2019, C2020, and constraint-based problems, should be included to demonstrate the effectiveness of ITLCO.
8. The images in this study are of low quality; the resolution should be increased to at least 330 ppi to meet the journal's standards.
9. The introduction should be enhanced with recent algorithms.

Reviewer 2 ·

Basic reporting

The paper introduces an Improved Termite Life Cycle Optimizer Algorithm (ITLCO) aimed at enhancing the convergence performance and robustness of the original Termite Life Cycle Optimizer (TLCO) for global optimization problems.Worker Generation Strategy enhances communication between individual termites within the worker population and the global optimal solution.Also, it balances convergence speed and population diversity, reducing the risk of getting trapped in local optima. Soldier Generation Strategy introduces a step factor based on evolutionary principles and promotes convergence speed by leveraging optimal or near-optimal individuals for generating solutions. Replacement Update Mechanism implements an adaptive crossover mechanism for updating populations.Balances convergence and diversity by ensuring suboptimal solutions do not replace superior individuals prematurely. However, following comments should be taken into considereation for revised paper.
- The paper is written in clear, making it accessible to the intended audience. However, a few minor grammatical improvements could enhance readability further. For example;"This work presents a enhanced( should be an enhanced) termite life cycle optimizer algorithm (ITLCO) with the aim of enhancing both the speed and accuracy of convergence." "This strategy aims to enhance (should include the) convergence speed while preserving population diversity."
- The authors provide a sufficient background on the termite-inspired optimization algorithm and situate their work within the broader context of metaheuristic optimization research. Relevant literature is referenced, but expanding the discussion on competing algorithms or related recent studies could strengthen the contextual framework.
- The results include terms, equations, and mechanisms. However, providing additional detailed explanations for some equations and their practical implications could improve accessibility for readers less familiar with the underlying principles.

Experimental design

The article aligns well with the aims and scope of the journal, presenting original primary research that is relevant and meaningful. The research question is clearly defined, and the authors effectively state how their work addresses a gap in the optimization algorithm literature. However, there are areas where improvements could strengthen the manuscript:
- The evaluation relies heavily on the CEC2013 test set. Including additional datasets or more diverse real-world examples could provide a more robust validation of the algorithm's generalizability.
-The manuscript does not address the computational overhead of the proposed improvements. Adding a discussion on computational complexity and runtime performance compared to baseline methods would provide a more comprehensive evaluation.
-Some sections, such as the replacement update mechanism, use highly technical language without sufficient explanation for its practical implications. Simplifying or providing illustrations could enhance understanding.

Validity of the findings

The article meets the standards in several areas, with clear linkage between the research question, methods, and conclusions. However, a few areas could be improved:

- A more explicit discussion on the scenarios or types of optimization problems where ITLCO would outperform existing methods would be beneficial.
- Although the underlying data and statistical tests are robust, the presentation of statistical results could be improved. For example, providing more context on the significance of the Wilcoxon rank-sum and Friedman tests and discussing the practical implications of these results would enhance the paper's clarity.
-The conclusions are well-aligned with the supporting results but could include more emphasis on practical implications, such as how the ITLCO can be adapted for real-world applications or its potential limitations.

---

## Round 0.2 · Minor Revisions

Dear Authors,

Thank you for revising your article. Feedback from the reviewers is now available. We strongly recommend that you address the issues raised by Reviewer 1, related to computational complexity and resubmit your paper after making the necessary additions and changes.

Best wishes

Reviewer 1 ·

Basic reporting

no comment

Experimental design

no comment

Validity of the findings

no comment

Additional comments

As mentioned in the first round, adding equations (10) and (11) will increase the computation time of ITLCO compared to the original algorithm. As shown in the results in Table 12, ITLCO takes more time than the original algorithm. Therefore, the current version of ITLCO is not meaningful. I cannot recommend this paper for acceptance unless the authors can improve the computation time of ITLCO.

I suggest another revision. If the authors are unable to improve the computation time of ITLCO, my final decision will be rejection

Reviewer 2 ·

Basic reporting

The revised paper demonstrates that the authors addressed concerns about insufficient context by adding new references to recent algorithms. The authors effectively addressed comments regarding figures, tables, and resolution issues. Enhancements like increasing figure resolution and providing a scatter plot for population diversity are commendable. The authors detail their additions, such as comparisons on new datasets (CEC2019, CEC2020) and an extended discussion of statistical tests, strengthening the linkage between hypotheses and findings.

Experimental design

The revised paper demonstrates that the research aligns with the journal's aims and addresses a meaningful knowledge gap in optimization algorithms. The authors provided rigorous investigation through additional datasets, statistical analyses, and detailed explanations of key equations, enhancing the study's robustness.

Validity of the findings

The revised paper effectively highlights the novelty of the ITLCO algorithm by addressing how its new strategies improve convergence and diversity in metaheuristic optimization. The underlying data and statistical analyses are robust, using established benchmarks (CEC2019, C2020) and tests like Wilcoxon rank-sum and Friedman analyses to validate results.

---

## Round 0.3 · accepted · Accept

Dear Authors,

Thank you for addressing the reviewers' all comments. Your paper seems sufficiently improved and ready for publication.

Best wishes,

Reviewer 1 ·

Basic reporting

no comment

Experimental design

no comment

Validity of the findings

no comment

Additional comments

no comment